# DNA-assisted swarm control in a biomolecular motor system

Jakia Jannat Keya[1], Ryuhei Suzuki[1], Arif Md. Rashedul Kabir[2], Daisuke Inoue[2], Hiroyuki Asanuma[3], Kazuki Sada[1,2], Henry Hess [4], Akinori Kuzuya [5] & Akira Kakugo[1,2,4]

In nature, swarming behavior has evolved repeatedly among motile organisms because it confers a variety of beneficial emergent properties. These include improved information gathering, protection from predators, and resource utilization. Some organisms, e.g., locusts, switch between solitary and swarm behavior in response to external stimuli. Aspects of swarming behavior have been demonstrated for motile supramolecular systems composed of biomolecular motors and cytoskeletal filaments, where cross-linkers induce large scale organization. The capabilities of such supramolecular systems may be further extended if the swarming behavior can be programmed and controlled. Here, we demonstrate that the swarming of DNA-functionalized microtubules (MTs) propelled by surface-adhered kinesin motors can be programmed and reversibly regulated by DNA signals. Emergent swarm behavior, such as translational and circular motion, can be selected by tuning the MT stiffness. Photoresponsive DNA containing azobenzene groups enables switching between solitary and swarm behavior in response to stimulation with visible or ultraviolet light.

[1] Graduate School of Chemical Sciences and Engineering, Hokkaido University, Sapporo, 060-0810, Japan. [2] Faculty of Science, Hokkaido University, Sapporo, 060-0810, Japan. [3] Graduate School of Engineering, Nagoya University, Osaka, 564-8680, Japan. [4] Department of Biomedical Engineering, Columbia University, 1210 Amsterdam Avenue, New York, NY, 10027, USA. [5] Department of Chemistry and Materials Engineering, Kansai University, Osaka, 564-8680, Japan. Correspondence and requests for materials should be addressed to A.K. (email: kuzuya@kansai-u.ac.jp) or to A.K. (email: kakugo@sci.hokudai.ac.jp)

Nature assembles its wide variety of structures through self-assembly and self-organization where local interactions among the individual components play pivotal roles[1]. A striking example is the swarming of living organisms in which fascinating, large scale and complex structures emerge from local interactions among the individuals rather than through control by a leader[2]. Fish schools, ant colonies, and bird flocks are typical examples of swarming observed in nature. Swarming grants several advantages to the organism, including parallelism, robustness, and flexibility; all of these are unachievable by a single entity[3,4]. Inspired by the attractive features of swarming organisms, researchers working in the fields of robotics and nanotechnology have been investigating the swarming of self-propelled mechanical devices[5–7] and chemically powered self-propelled particles[8–10], respectively. A major challenge in swarm robotics is the construction of large numbers of individual robots capable of programmable self-assembly[11].

This challenge may be addressed by the creation of molecular robots. Molecular robots are molecular systems composed of all of the three essential components of robots: sensors, information processors, and actuators[12–14]. The systems integrating biomolecular motors and their corresponding cytoskeletal filaments, such as myosin–actin, dynein-MT, or kinesin-MT systems, can provide a large number of self-propelled molecular entities[15,16]. Recent studies have demonstrated some aspects of swarming behavior by controlling the mutual interactions of the propelled filaments using associated proteins[17], depletion agents[18], crowding effects[19–21], or ligand–receptor-based crosslinking[22–25]. These interactions lead to the emergence of fascinating swarm patterns, such as bundles, spools, vortex lattices[26,27], as well as circular or polar patterns[28]. However, programmability of the interactions, critical to the exploration of swarm behavior at the macroscale[11], has not yet been achieved in these systems. Therefore, the scientific questions are if a linker exists that combines sufficient interaction strength, selectivity, reversibility, and controllability to enable these systems to respond with distinct behaviors, and if that linker can provide sensing and information processing capabilities to these systems composed of cytoskeletal filaments propelled by biomolecular motors.

We demonstrate, in this study, that DNA can be employed as a universal interface to control the swarming of MTs in a programmable manner. The unique features of DNA as a storage device of genetic information, i.e., strict sequence-selectivity managed by complementary base-pairing and uniform right-handed double helical structure, together with recent progress in chemical DNA synthesis, have made DNA a versatile tool for molecular computing[29,30], and a building block for the construction of nanostructures[31,32]. DNA has already been used in combination with biomolecular motor systems, either acting as a highly specific glue to assemble motors into multimers[33,34], or to connect motors to DNA origami scaffolds[35,36], or being conjugated to MTs for the purpose of cargo loading/unloading[37–45]. Here, we report the control of swarming of kinesin-propelled MTs (actuators) by tethering single-stranded DNA to the MTs and programming the interactions among MTs using DNA crosslinkers as input signals (information processing). We also regulated the swarming mode of the MTs, such as swarming with translational or circular motion, by tuning the physical properties of the MTs. By introducing a photoresponsive residue, azobenzene, into the DNA strands as a sensor, the swarming of the MTs was further controlled by photoirradiation in a reversible fashion.

## Results

### Design of swarming of MTs controlled by DNA.
We prepared the individual swarm units by conjugating MTs with single-stranded DNA, at a labeling ratio of about one strand per tubulin dimer, through a copper-free click reaction. Either the DNA or the MTs were labeled with a fluorescent dye in order to allow monitoring of the MTs under a fluorescence microscope (Fig. 1a). The MTs are cylindrical objects with an outer diameter of 25 nm and lengths between 2 and 10 μm. Based on the results of melting temperature ($T_m$) simulation, the base number in a DNA strand was fixed to 16 to obtain a $T_m$ above the working temperature of the biomolecular motor system (25 °C). The DNA-conjugated MTs were propelled by surface-adhered kinesins using the chemical energy of adenosine triphosphate (ATP)[46]. Smooth gliding of the DNA-conjugated MTs confirmed that the DNA conjugation does not hinder the interaction of MTs with kinesins (Supplementary Figs. 1 and 2 and Supplementary Tables 1 and 2).

### Demonstration of reversible swarming of MTs triggered by DNA.
Two DNA strands, $T_{16}$ and $(TTG)_5$, were labeled with the fluorescent dyes TAMRA (red) and FAM (green), respectively. These strands, termed receptor DNA (r-DNA), were conjugated to MTs, yielding two types of MTs distinguishable by their fluorescence (Fig. 1b and Supplementary Table 3). We placed equal numbers of red ($T_{16}$-labeled) and green ($(TTG)_5$-labeled) MTs on a kinesin-coated substrate (Fig. 1c) at a combined density of 50,000 mm$^{-2}$. In the presence of ATP, the MTs moved without any interaction with each other (Fig. 1d and Supplementary Movie 1). Linker DNA (l-DNA1) was designed to be partially complementary to the r-DNAs, so that it can cross-link them (Fig. 1e and Supplementary Table 3). Association of the MTs was then initiated by the introduction of l-DNA1 as the input signal (Fig. 1e–g). While gliding, the red and green MTs came close to each other, associated into swarms (appearing yellow in the merged images), and continued moving (Fig. 1g). The size of the swarms of MTs increased by mergers of swarms, decreasing the density of swarms over time (Fig. 1g and Supplementary Movie 1). Despite the increase in size, the swarms of MTs exhibited translational motion with a velocity (0.51 ± 0.02 μm s$^{-1}$) close to that of individual MTs (0.60 ± 0.05 μm s$^{-1}$). We counted the number of individual MTs at different time points and characterized the swarming of MTs by calculating the association ratio defined as the fraction of the number of initial MTs incorporated into the swarms (see Supplementary Methods). The association ratio increased with time and reached a plateau (~90–95%) within 60 min after the addition of l-DNA1 (Fig. 1i). In the absence of kinesin, freely diffusing MTs formed unstructured aggregates whereas immobile MTs did not interact in the presence of kinesin and absence of ATP (Supplementary Fig. 3).

Dissociation of the swarms into individual MTs was then demonstrated by introducing dissociation DNA (d-DNA), which was designed to extract l-DNA1 through a DNA strand exchange reaction (Figs. 1e, f and Supplementary Table 3)[47]. The yellow-colored swarms of MTs dissociated into red and green MTs after the introduction of d-DNA (Figs. 1h, i). By counting the individual MTs present after dissociation (Fig. 1i and Supplementary Movie 1), it was estimated that a swarm is composed of ~100 individual MTs.

### DNA-based logic gates to control the swarming of MTs.
Based on the utility of DNA as an operator for molecular computing[30], we aimed to demonstrate different logic operations where swarming was the output controlled by DNA inputs (Supplementary Table 4). A YES logic gate was realized by using l-DNA1 as input 1, and the swarming of red and green MTs as the output 1 (Fig. 2). The AND logic gate was demonstrated by designing two different linker DNA signals as l-DNA2 and

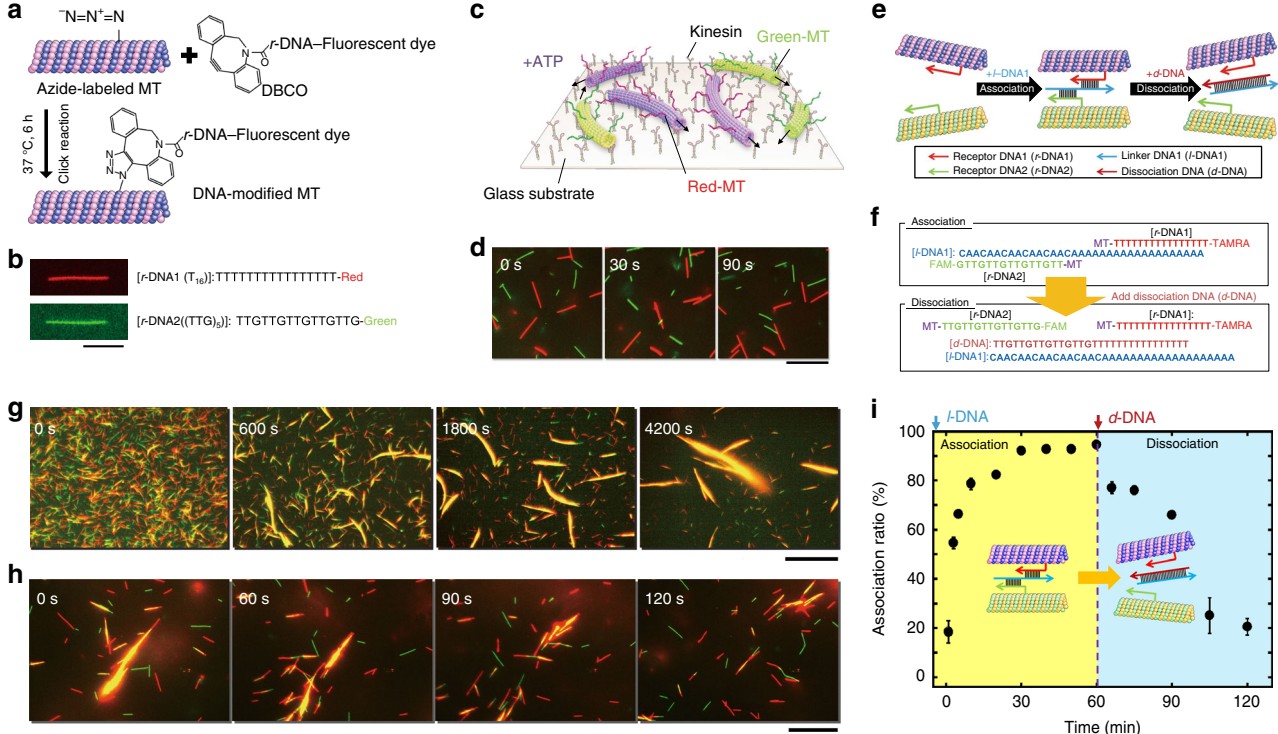

**Fig. 1** Preparation of MTs and control of their swarming. **a** Conjugation of $r$-DNA to azide-functionalized MTs by a click reaction. **b** TAMRA (red) and FAM (green) labeled DNA-conjugated MTs with sequences of $T_{16}$ ($r$-DNA1) and $(TTG)_5$ ($r$-DNA2), respectively. Scale bar: 5 μm. **c** Schematic of red and green MTs gliding on kinesins. **d** Time-lapse images showing motility of rigid MTs with lengths of $5 \pm 2$ and $5 \pm 3$ μm (average ± s.d.) conjugated to $T_{16}$ and $(TTG)_5$, respectively. Scale bar: 10 μm. **e** Schematic of the association of red and green MTs by $l$-DNA1 and their dissociation by $d$-DNA via extraction of $l$-DNA1 through a DNA strand exchange reaction. **f** Association and dissociation of MTs by DNA base pair interaction. **g** Time-lapse images showing swarming of MTs. **h** Dissociation of swarms of MTs. Scale bar: 20 μm (**g** and **h**). **i** Change of association ratio over time after introduction of $l$-DNA1 (0.6 μM) and $d$-DNA (0.6 μM). The concentration of the red and green MTs was 0.6 μM each, and the kinesin concentration was 0.3 μM. Error bar: standard error (s.e.m.)

$l$-DNA3, which are partially complementary to $r$-DNA1 and $r$-DNA2, respectively, and also to each other. The OR logic gate was implemented by conjugating pairs of $r$-DNA to the MTs ($r$-DNA1 and $r$-DNA3 labeled with TAMRA; $r$-DNA2 and $r$-DNA4 labeled with FAM), and using $l$-DNA1 (complementary to $r$-DNA1 and $r$-DNA2) and $l$-DNA4 (complementary to $r$-DNA3 and $r$-DNA4) as the two input signals. Association ratios of 85–100% were obtained for all of the systems expecting the output representing 1 (Fig. 2), which are significantly higher than those for the outputs representing 0 (<5%).

**Swarming modes regulated by the physical properties of MTs.**
We have previously found that the swarming mode of MTs is controlled by the rigidity and length of the MTs[48,49]. We prepared MTs with lower rigidity by polymerizing MTs with guanosine triphosphate (GTP) guanylyl-(α,β)-methylenedisphosphonate (GMPCPP) used in the experiments described above[50]. The flexible GTP-MTs were then conjugated with $r$-DNA1 and $r$-DNA2. While gliding on kinesins, the flexible MTs moved in more curved paths compared to the rigid MTs (Fig. 3a, b, Supplementary Movie 2). The path persistence length, $L_\mathrm{p}$, of the flexible GTP-MTs was $245 \pm 32$ μm (Supplementary Fig. 4), while that of the rigid GMPCPP-MTs was $582 \pm 97$ μm, reflecting a more than two-fold difference in rigidity. Unlike the rigid MTs, the flexible MTs exhibited swarming with circular motion upon the input of $l$-DNA1 (Fig. 3c, d, and Supplementary Movie 2). The swarms with circular motion can also be dissociated into single MTs by introducing the input $d$-DNA signal (Fig. 3e, f, and

Supplementary Movie 2). Single MTs retained their gliding motion on the kinesin-coated substrate.

**Orthogonal control of swarming of MTs.** The selective hybridization property of DNA allowed us to design a system exhibiting controlled swarming of flexible and rigid MTs without any crosstalk. We conjugated two types of MTs, which differed in body length and rigidity, with two different DNA logic gates (Fig. 4 and Supplementary Table 5). The flexible MTs associated into swarms with a circular motion in the presence of $l$-DNA1, while the swarms of the rigid MTs exhibited a translational motion in the presence of $l$-DNA5 (Supplementary Movie 3). The formation of these two types of swarms was completely orthogonal, because $l$-DNA1 did not affect the rigid MTs while $l$-DNA5 did not interact with the flexible MTs (Fig. 4). The independent addressability of the MTs based on multiple logic gates will allow the design of more complex systems with diverse functionality.

**Light-switched repeated swarming of MTs.** To obtain a fast, reversible, and non-invasive way of altering the DNA input, we aimed to incorporate photoresponsive DNAs ($p$-DNAs)[51]. We installed a photoresponsive molecule, azobenzene, into two DNA strands which allowed the ON/OFF switching of the hybridization between these two DNA strands. The switching arises from melting temperature ($T_\mathrm{m}$) changes of DNA hybridization triggered by the cis–trans isomerization of the azobenzene moiety by ultraviolet (UV) or visible light (Fig. 5a and Supplementary Table 6)[52]. The two $p$-DNAs were designed such that the melting

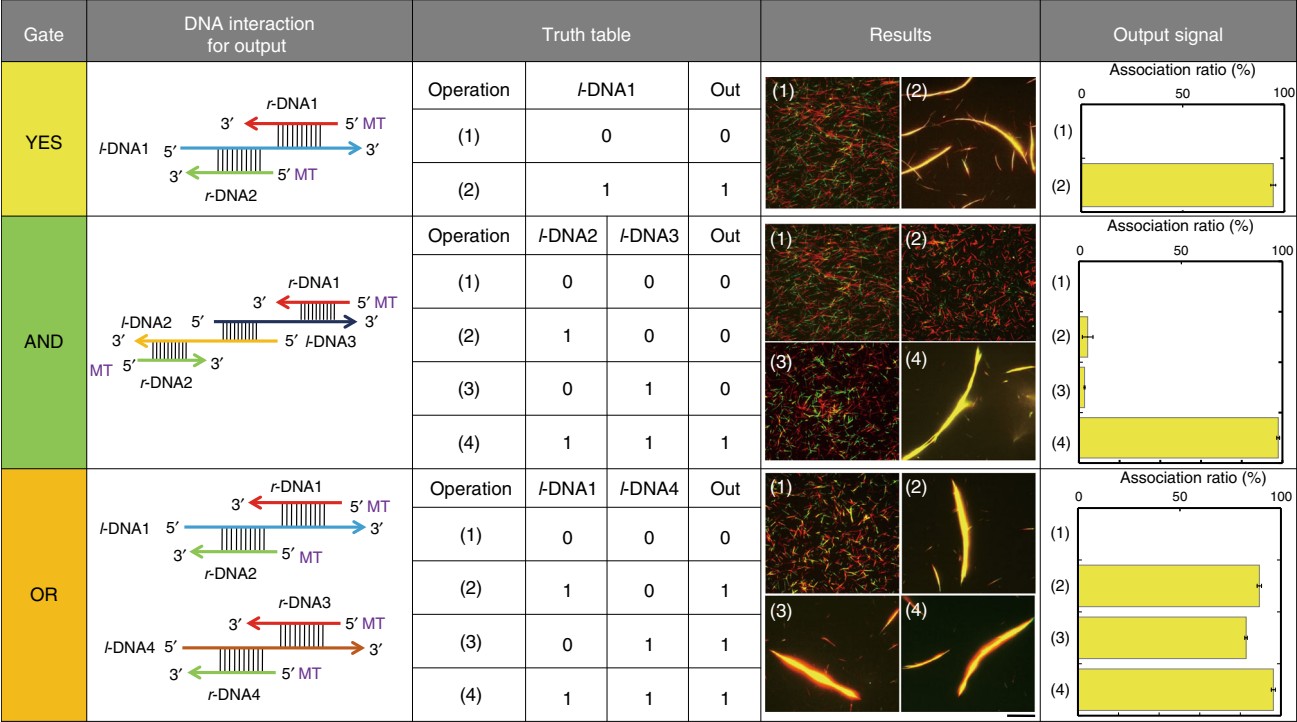

**Fig. 2** Design of logic gates constructed with MTs. For the YES gate, the *l*-DNA1 signal was inputted into the system and swarming was obtained as the output signal (1 to 1). For the AND gate, *l*-DNA2 and *l*-DNA3 had both to be present to obtain swarming. For the OR gate, the presence of either *l*-DNA1 or *l*-DNA4 was sufficient to obtain swarming. The concentration of red and green MTs was 0.6 μM, and the conjugation ratio of each *r*-DNA to MTs was ~100%. The concentration of kinesin and each *l*-DNA was 0.3 μM and 0.6 μM, respectively. Scale bar: 20 μm. Error bars: s.e.m.

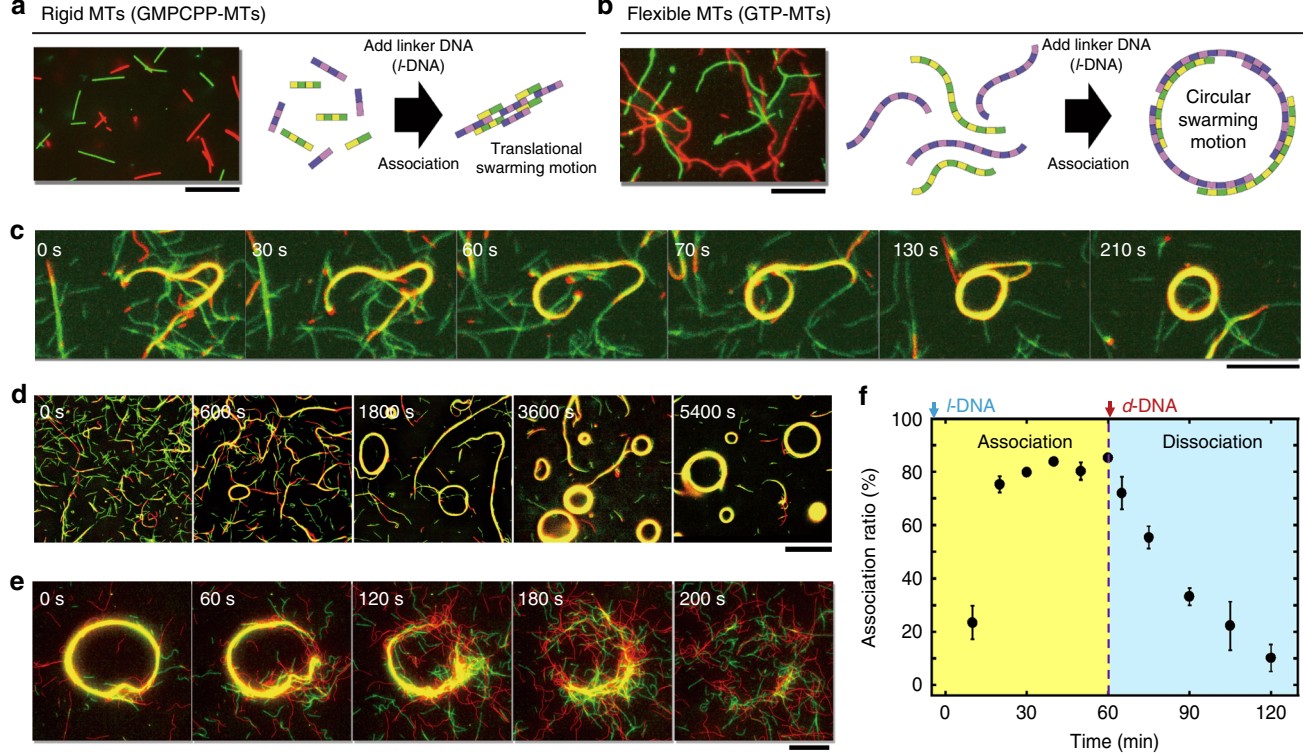

**Fig. 3** Regulation of swarming mode of MTs by tuning their physical properties. **a**, **b** Fluorescence microscopy images of rigid and flexible MTs, respectively; schematic illustrations showing different swarming modes of the MTs. **c** Time-lapse fluorescence microscopy images showing the formation of a swarm group with a circular motion from flexible red and green MTs with lengths of 22 ± 13 μm and 16 ± 9 μm (average ± s.d.), respectively. **d** Time-lapse fluorescence microscopy images showing the formation of swarms with a circular motion in the presence of *l*-DNA1 (0.6 μM). **e** Dissociation of a swarm group with a circular motion triggered by the *d*-DNA input signal (0.6 μM). The swarm group was made of ~300 single MTs. **f** Change of the association ratio over time upon addition of *l*-DNA1 and *d*-DNA. The concentration of the red and green MTs was 0.6 μM each, and the conjugation ratio of *r*-DNA1 or *r*-DNA2 to MTs was ~100%. The concentration of kinesin was 0.3 μM. Scale bar: 20 μm (**a**–**e**). Error bar: s.e.m

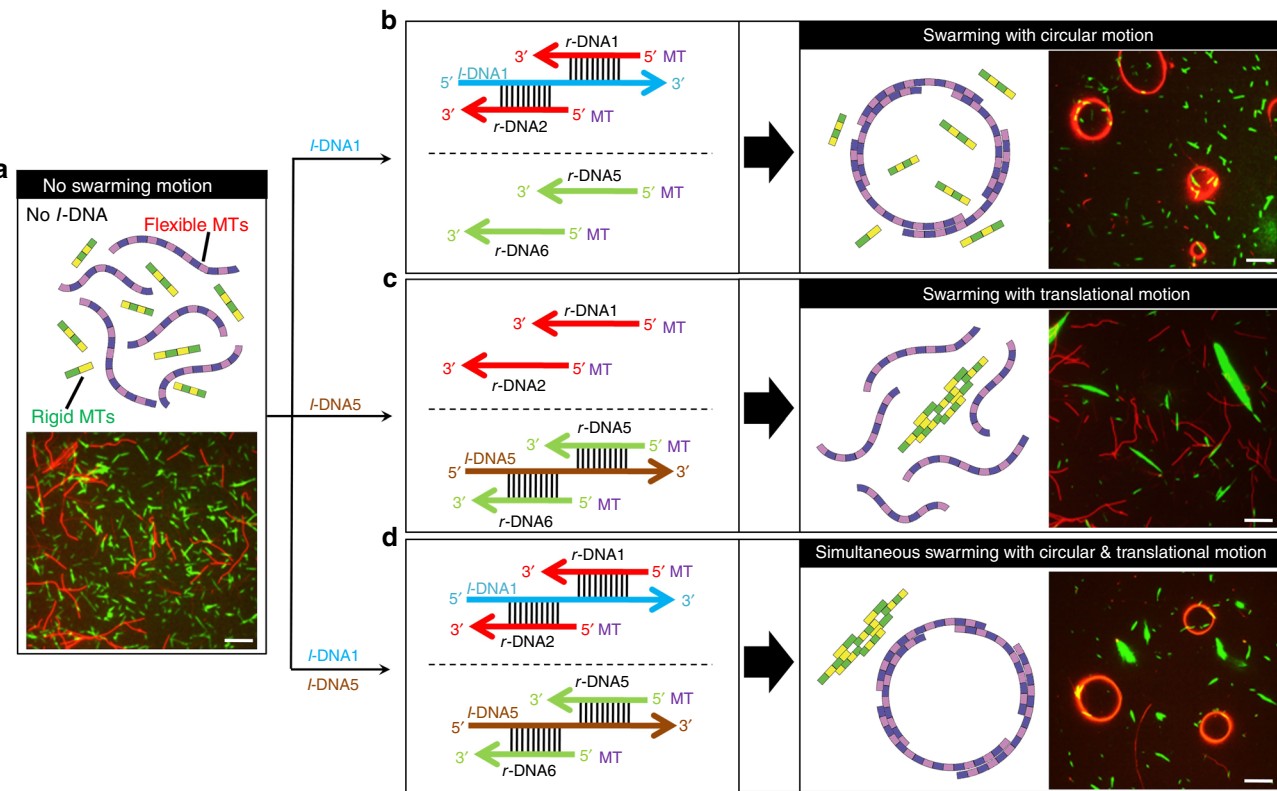

**Fig. 4** Orthogonal control of swarming of MTs. **a** Schematic representation and fluorescence microscopy image of MTs with different rigidity. The flexible MTs (red) were conjugated with *r*-DNA1 and *r*-DNA2, while the rigid MTs (green) were conjugated with *r*-DNA5 and *r*-DNA6. **b** Upon inputting *l*-DNA1, the flexible MTs associated into circular shaped swarms through hybridization of *r*-DNA1 and *r*-DNA2 with *l*-DNA1 and appeared in red. **c** Green swarms with translational motion associated with rigid MTs were formed through hybridization of *r*-DNA5 and *r*-DNA6 with *l*-DNA5. **d** Swarms with translational and circular motions were simultaneously formed in response to the introduction of both input DNA signals. The concentration of red and green MTs was 0.6 μM each, and the conjugation ratio of each *r*-DNA to MTs was ~100%. The concentration of kinesin was 0.3 μM and that of each *l*-DNA was 0.6 μM. Scale bars: 20 μm

temperature is <20 °C in the *cis* state and 60 °C in the *trans* state (Supplementary Fig. 5). The photoresponsive DNAs (*p*-DNA1 and *p*-DNA2) were conjugated to MTs already fluorescently labeled with TAMRA and FAM (Fig. 5b). The functionalized MTs moved on a kinesin-coated substrate without the loss of mobility (Supplementary Fig. 6). We applied UV irradiation ($\lambda = 365$ nm) to initialize the azobenzene groups to the *cis* form, in order to start with the movement of isolated MTs (Fig. 5c). We then irradiated the MTs with visible light ($\lambda = 480$ nm), which triggered the *cis* to *trans* isomerisation of azobenzenes, permitted the hybridization of *p*-DNA1 with *p*-DNA2, and caused the swarming of MTs (Fig. 5c, rigid MTs). Subsequent irradiation with UV dissociated the swarms back into individual MTs. This light-switched association/dissociation of MTs was repeated for three cycles (Fig. 5d). We also changed the swarming mode of photoresponsive MTs from translational to circular motion by reducing the MT rigidity (Fig. 5c, flexible MTs). Repetitive and reversible switching of the swarming of MTs with light was thus successfully achieved by installing a photoresponsive component to the system (Supplementary Movie 4).

## Discussion

In response to the scientific questions, we demonstrated that DNA can be employed to selectively and reversibly control the behaviors of cytoskeletal filaments propelled by biomolecular motors and to provide sensing and information processing capabilities to these systems. Our approach complements the recently described approach to information processing where a swarm of cytoskeletal filaments propelled by biomolecular motors traverses a maze of guiding structures and in the process, computes the solution to a mathematical problem encoded in the maze[53]. Here, the information is not encoded in the positions of the filaments with respect to a guiding structure, but in their positions relative to each other. This work should motivate further advancement not only in chemistry, but also in bio- and DNA nanotechnology. Potential future applications are: active DNA sensors controlled by analyte oligonucleotides such as microRNA, biomimetic displays where DNA computing produces patterns, adaptive actuators designed to sense and respond to their chemical and/or physical environment, analyte concentrators which integrate preprocessing of molecular information, and sequential reactors with programmable reaction pathways. At the same time, our work capitalizes on the advantages of biomolecular motor systems, such as high energy efficiency, high specific power, and cost efficiency in handling millions of MTs, which are not yet available in conventionally fabricated mechanical swarm robots. The present work thus contributes to a new paradigm in robotics, i.e., molecular robotics.

## Methods

**Purification of tubulin and kinesin**. Tubulin was purified from porcine brain using a high-concentration PIPES buffer (1 M PIPES, 20 mM EGTA, and 10 mM MgCl₂) and stored in BRB80 buffer (80 mM PIPES, 1 mM EGTA, 2 mM MgCl₂, pH adjusted to 6.8 using KOH)[54]. Recombinant kinesin-1 consisting of the first 573 amino-acid residues of human kinesin-1 was prepared as described in the

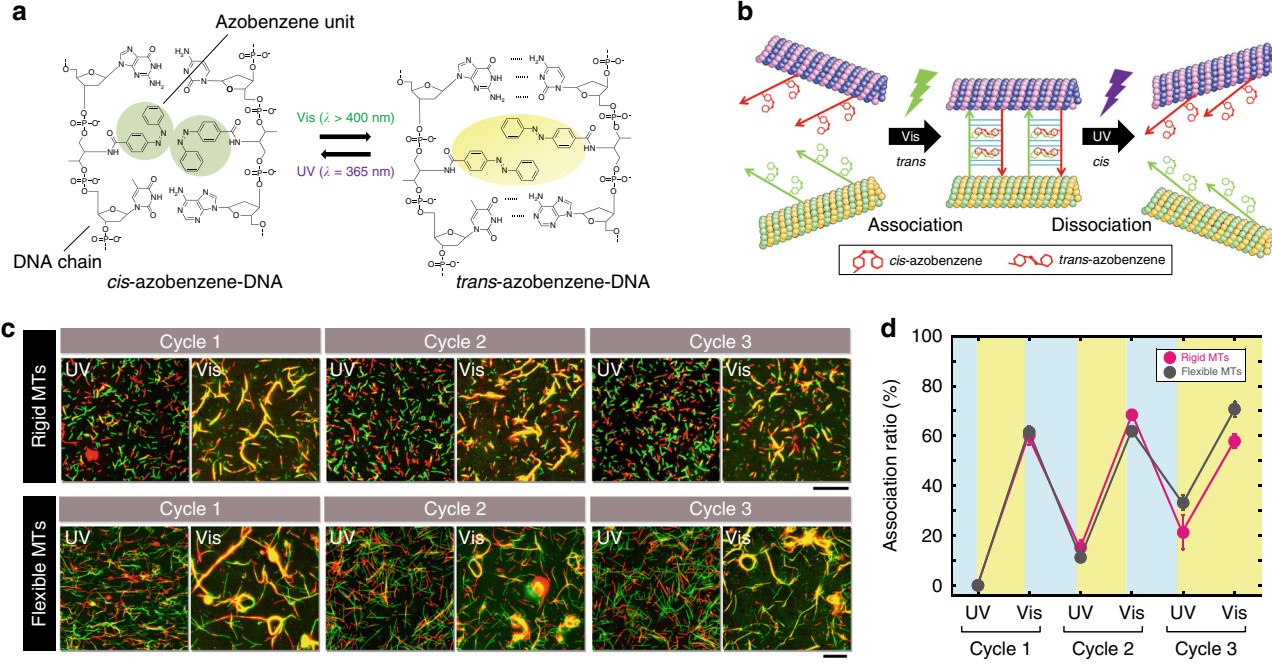

**Fig. 5** Light-switched, repeated swarming of MTs. **a** Reversible hydrogen bonding of photoresponsive DNA (*p*-DNA) by light-induced *cis*–*trans* isomerization of azobenzene. **b** Schematic of selective association and dissociation of *p*-DNA-conjugated MTs under UV and visible light irradiation, respectively. **c** Visible light ($\lambda = 480$ nm, $I = 1.3$ mW cm$^{-2}$) induced isomerization of azobenzene from the *cis* to the *trans* form, which resulted in translational swarming of the *p*-DNA-conjugated rigid MTs with length of $4 \pm 2$ μm (average ± s.d.). The swarms were then exposed to UV light ($\lambda = 365$ nm, $I = 0.4$ mW cm$^{-2}$) for 6 min that isomerized the azobenzene from the *trans* to the *cis* form. The swarms dissociated into single MTs within 12 min of the onset of photoirradiation. This cycle was repeated three times. Visible light irradiation to flexible MTs with length of $12 \pm 1$ μm (average ± s.d.) generated swarms with circular motion. **d** Changes in the association ratio upon repeated irradiation by visible and UV light. The concentration of red and green MTs was 0.6 μM each, and the conjugation ratio of *p*-DNA1 or *p*-DNA2 to MTs was ~100%. The concentration of kinesin was 0.8 μM. Scale bars: 20 μm. Error bar: s.e.m.

literature[55]. Azide labeled tubulin was prepared using N$_3$-PEG4-NHS following the established protocol of labeling tubulin with a fluorescent dye[56]. The tubulin concentration was determined by measuring the absorbance at 280 nm using a UV spectrophotometer (Nanodrop 2000c).

**Design and preparation of DNA sequences**. *r*-DNA and *l*-DNA strands were designed from $T_m$ simulation using OligoAnalyzer 3.1 (https://sg.idtdna.com/calc/analyzer) software with a $T_m$ between 0 and 50 °C for experimental testing. A further selection criterion was followed for logic gate experiments such that any undesired interactions were avoided between DNA strands (Supplementary Fig. 7). Dibenzocyclooctyne (DBCO) and fluorescent dye-labeled strands were chemically synthesized using appropriate CPG columns and a phosphoramidite (Glen Research, VA) on a ABI 3900 automatic DNA synthesizer, purified by reverse phase HPLC and fully characterized by MALDI-TOF/MS (Bruker microflex LRF). The *r*-DNA was modified at the 3′ end with either 5(6)-Carboxytetramethylrhodamine (TAMRA) or 5-Carboxyfluorescein (FAM) and at the 5′ end with DBCO. The *p*-DNA was synthesized in the laboratory according to an established protocol[57]. *l*-DNA and *d*-DNA were purchased from Eurofins Genomics LLC.

**Preparation of MTs**. MTs were prepared by adding azide tubulin to polymerization buffer (80 mM PIPES, 1 mM EGTA, 1 mM MgCl$_2$, 1 mM polymerizing agent, pH ~6.8) into a final concentration of 56 μM tubulin incubating at 37 °C for 30 min. The polymerizing agent for flexible MTs was GTP, and for rigid MTs was GMPCPP, a slowly hydrolyzable analog of GTP. Dimethyl sulfoxide (DMSO) was added to a final concentration of 5%, only for the polymerization of flexible MTs. Copper free click reaction was initiated by adding 3.5 μL DBCO conjugated *r*-DNAs (500 μM) to the 5 μL azide-MTs (56 μM) which allowed azide-alkyne cycloaddition reaction and incubated at 37 °C for 6 h[58]. 100 μL of cushion buffer (BRB80 buffer supplemented with 60% glycerol) was used to separate the MTs by centrifugation at 201,000 × *g* (S55A2-0156 rotor, Hitachi) for 1 h at 37 °C. After removing the supernatant, the pellet of *r*-DNA-conjugated MTs was washed once with 100 μL BRB80P (BRB80 supplemented with 1 mM taxol) and dissolved in 15 μL BRB80P. *p*-DNA-conjugated MTs were prepared following the same procedure.

**Demonstration of swarming of MTs**. A flow cell with dimensions of $9 \times 2.5 \times 0.45$ mm$^3$ (L × W × H) was assembled from two cover glasses (MATSUNAMI Inc.) using a double-sided tape as a spacer. The flow cell was filled with 5 μL casein buffer (BRB80 buffer supplemented with 0.5 mg mL$^{-1}$ casein). After incubating for 3 min, 0.3 μM kinesin solution was introduced into the flow cell and incubated for 5 min resulting in a kinesin density of 4000 μm$^{-2}$ on the substrate[59]. After washing the flow cell with 5 μL of wash buffer (BRB80 buffer supplemented with 1 mM DTT, 10 μM taxol), 5 μL of red MTs (TAMRA-labeled *r*-DNA MTs) solution was introduced and incubated for 2 min, followed by washing with 10 μL of wash buffer. Subsequently, 5 μL of green MTs (FAM-labeled *r*-DNA MTs) solution was introduced and incubated for 2 min, followed by washing with 10 μL of motility buffer. The green MTs were incubated with *l*-DNA for 15 min at room temperature prior to addition to flow cell. The motility of MTs was initiated by applying 5 μL ATP buffer (wash buffer supplemented with 5 mM ATP, 4.5 mg mL$^{-1}$ D-glucose, 50 U mL$^{-1}$ glucose oxidase, 50 U mL$^{-1}$ catalase, and 0.2% methylcellulose (w/v)). The time of ATP addition was set as 0 h. Soon after the addition of ATP buffer, the flow cell was placed in an inert chamber system (ICS)[60] and the MTs were monitored using a microscope at room temperature (25 °C). The experiment was performed at least 10 times for each condition.

**Fluorescence microscopy**. The samples were illuminated with a 100 W mercury lamp and visualized by an epifluorescence microscope (Eclipse Ti, Nikon) using an oil-coupled Plan Apo 60× N.A.1.4 objective (Nikon). UV cut-off filter blocks (TRITC: EX 540/25, DM565, BA605/55; GFP-B: EX460-500, DM505, BA510-560; Nikon) were used in the optical path of the microscope. Images were captured using a cooled-CMOS camera (NEO sCMOS, Andor) connected to a PC. Two ND filters (ND4, 25% transmittance for TRITC and ND1, 100% transmittance for GFP-B) were inserted into the illumination light path of the fluorescence microscope to reduce photobleaching of the samples. In order to isomerize the azobenzene units, the flow cell was irradiated with the light passed through a UV-1A filter block (UV-1A: EX 365-410, DM400, BA400; Nikon).

**Data availability**. The data that support this study are available from the corresponding author upon reasonable request.

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

## Acknowledgements

We thank Prof. Akihiko Konagaya and the members of the Molecular Robotics Research Group for valuable discussions. This work was financially supported by Grant-in-Aid for Scientific Research on Innovative Areas "Molecular Robotics" (JSPS KAKENHI Grant

Number JP24104004) from Japan Society for the Promotion of Science (JSPS) and Grant-in-Aid for Challenging Exploratory Research (JSPS KAKENHI Grant Number 15K12135). H.H. was supported by NSF grant CMMI-1662329.

## Author contributions

A.M.R.K., D.I., K.S., A.Ku., and A.Ka. conceived and designed the experiments. H.A. synthesized the azobenzene-amidite monomer. J.J.K. and R.S. performed the experiments, and analyzed the experimental results. J.J.K., A.M.R.K., D.I., K.S., H.H., A.Ku., and A.Ka. wrote the manuscript.

## Additional information

**Competing interests:** The authors declare no competing financial interests.

