## [Peer Review File · Nature Communications]

Reviewers' comments:

Reviewer #3 (Remarks to the Author):

I have read the re-submitted article, and the rebuttal, with great care, and my overall conclusion is that the paper improved, but rather progressing asymptotically towards a threshold that would justify its publication in Nature Communications. The main advance comes from the clarification of the concept of "robots", which was a contentious issue for all reviewers. This being said, I would still argue that the "information processing" does not occur - DNA-DNA recognition is not information processing (I can agree with the other two parameters of the robots). All other improvements are quite small, and mostly about semantics (or references).

My assessment stands as it was.

Reviewer #4 (Remarks to the Author):

This paper by Keya et al, reports DNA-assisted swarm control in a system where gliding microtubules are driven by surface-adsorbed kinesin motors. The on/off switching of microtubule swarming is programmed and reversibly regulated by DNA and light signals. Further, by varying the mechanical properties (persistence length) of the microtubules, either circular or translational motion is observed during swarming.

The paper is largely well and clearly written with illustrative figures. The methods are described in sufficient detail and the experiments seem to be performed to high standard.

In accordance with the rebuttal letter by the authors, in response to my comments on their submission to Nature Chemistry, the main novelty of the paper lies in the combined use of a range of previously developed techniques. The previous papers as a basis for these techniques are now fully cited. The novel combination of techniques opens for new applications and possible insights. A weakness is that the authors, only to a limited extent, specify direct scientific questions as basis for their studies. As a corollary, the analysis and discussion are quite superficial and the most important insights are not very clear to a reader. Primarily the paper explores how different factors affect the swarming behavior without strong efforts to gain detailed insights. However, having stated this, the explorations certainly lead to interesting findings that may be of general value in the field and with potential to influence future thinking. I also think that the claims made, are convincing and partly

appropriately discussed in relation to previous results in the literature. Also the statistical analysis is sound. Therefore, considering the guiding questions for the review, I am largely in favor of publication after appropriate revision as suggested below.

With regard to the lack of specific scientific questions and detailed analysis, I feel that improvements are possible to strengthen the paper and justify its publication. For instance, it may be possible to reformulate some parts of the Introduction and the Discussion in terms of specific scientific questions and the addressing of these. Such reformulations may help the reader to more easily grasp the central elements of the study e.g. as a basis for future developments of applications and/or with regard to insights into the operation of self-assembled “molecular robots in the cell and elsewhere. It would also be desirable that more details are specified to clarify how some of the numerous examples of applications mentioned in the Conclusions may be achieved based on the described methods for swarming control.

Further minor improvements are suggested below:

1. Appreciable shortening of the Supplementary Material. Particularly, please filter out a substantial number of the supporting figures only leaving those that are most important. This will help a reader to grasp which findings are really central.

2. Changes to the text

-p4, lines 61-63 ““swarming” in a highly programmable manner. We demonstrate, in this study, that DNA can be employed as a universal interface to control swarming of MTs in a highly programmable manner..” I suggest that the authors reformulate this, e.g. remove “highly” and avoid repeating “programmable manner”.

-p. 6, line.126. “..output controlled..” I guess, “..output was controlled..”

Reviewer #5 (Remarks to the Author):

In the present manuscript the authors present that the swarming behavior of microtubules can be controlled by DNA hybridization. They put the ability of switching the interactions into a framework of robotics and computing. This is a rather unusual perspective – and somehow lacks the punch to really convince. I find the introduction a bit disturbingly full with buzz words.

The results are kind of interesting – indeed by controlling the hybridization of DNA- functionalized MTs different dynamic structures are formed – reminiscent of crosslinkers in actin motility assay (e.g Schaller et al, Soft Matter 2013). Moreover a light switching of DNA hybridization was used to control the dynamic association. The results are nice and convincing.

Outline of Revision

Answers to the comments from Reviewer #3:

Comment 1: I have read the re-submitted article, and the rebuttal, with great care, and my overall conclusion is that the paper improved, but rather progressing asymptotically towards a threshold that would justify its publication in Nature Communications. The main advance comes from the clarification of the concept of "robots", which was a contentious issue for all reviewers. This being said, I would still argue that the "information processing" does not occur - DNA-DNA recognition is not information processing (I can agree with the other two parameters of the robots). All other improvements are quite small, and mostly about semantics (or references). My assessment stands as it was.

Response: Taking into account the overall reviewers' comments, we perceive that the results presented in the manuscript are of interest to them. The reviewer (#3), however, may not yet be fully convinced of the concept of "molecular robotics". We agree that the idea seems "unusual" at this moment for researchers from the "wet and soft-matter" scientific communities. However, "molecular robotics" is emerging as an interdisciplinary field where also engineers with an interest in dynamics and control, for example, contribute. Moreover, a few studies based on similar concepts have recently appeared in other prestigious journals: i.e. the study pointed out by reviewer #3 himself or herself (ref 14 – Kassem et al.: Stereodivergent synthesis with a programmable molecular machine. *Nature* **549**, 374 (2017)), or ref. 13 (Sato et al.: Micrometer-sized molecular robot changes its shape in response to signal molecules. *Science Robotics* **2**, eaal3735 (2017)). We believe that the originality of the present study lies in the employment of literally dynamic "molecular actuators" for the first time, which enabled us to demonstrate "swarming" in a molecular system. This point is the key reason that qualifies our study to be published in Nature Communications, and we are confident that it will be appreciated by the research community in this emerging field.

As for the "information processing" argument, we would like to point out that "nature.com" has its own individual subject site entitled "DNA computing (<https://www.nature.com/subjects/dna-computing>)" with a definition: "DNA computing is a branch of biomolecular computing concerned with the use of DNA as a carrier of information to make arithmetic and logic operations." We believe that, regardless of the complexity of the operation, DNA-DNA recognition that precisely discriminates artificially designed complementary sequences is the basic information processing step on which this well-established research field is founded. This issue may be seen in the context of a recent publication (ref. 53 – Nicolau DV Jr et al.: "Parallel computation with molecular-motor-propelled agents in

nanofabricated networks”, PNAS 113, 2591 (2016)), where information processing is achieved by letting cytoskeletal filaments interact with a guiding structure encoding a mathematical problem. In Nicolau DV Jr et al., information is encoded in the positions of the cytoskeletal filaments relative to the guiding structure, in our work, information is encoded in the positions of the cytoskeletal filaments with respect to each other. Both approaches have distinctive advantages and complement each other. We added a discussion of this point to the conclusion and also the reference.

Answers to the comments from Reviewer #4:

Comment 1: With regard to the lack of specific scientific questions and detailed analysis, I feel that improvements are possible to strengthen the paper and justify its publication. For instance, it may be possible to reformulate some parts of the Introduction and the Discussion in terms of specific scientific questions and the addressing of these.

Response: We added the following sentences to the Introduction part to clarify the major scientific finding and significance of the present study as suggested: " Therefore, the scientific questions are if a linker exists that combines sufficient interaction strength, selectivity, reversibility, and controllability to enable these systems to respond with distinct behaviors, and if that linker can provide sensing and information processing capabilities to these systems composed of cytoskeletal filaments propelled by biomolecular motors. " in the end of second paragraph of the Introduction and to answer to these questions we added the following sentence: “In response to the scientific questions, we demonstrated that DNA can be employed to selectively and reversibly control the behaviors of cytoskeletal filaments propelled by biomolecular motors and to provide sensing and information processing capabilities to these systems.” in the Conclusion.

Comment 2: 1. It would also be desirable that more details are specified to clarify how some of the numerous examples of applications mentioned in the Conclusions may be achieved based on the described methods for swarming control.

Response: We have modified the sentence describing possible applications as follows: " Potential future applications are: active DNA sensors controlled by analyte oligonucleotides such as microRNA, biomimetic displays where DNA computing produces patterns, adaptive actuators designed to sense and respond to their chemical and/or physical environment, analyte concentrators which integrate pre-processing of molecular information, and sequential reactors with programmable reaction pathways."

Comment 3: Appreciable shortening of the Supplementary Material. Particularly, please filter out a substantial number of the supporting figures only leaving those that are most important. This will help a reader to grasp which findings are really central.

Response: We have removed Figure S3, S4, S5, S6, S7, S9, S11, S12, S13, S14, S15 and S16 from the revised Supporting Information, all of which are not required for the discussion in the main text.

Comment 4: -p4, lines 61-63 “"swarming" in a highly programmable manner. We demonstrate, in this study, that DNA can be employed as a universal interface to control swarming of MTs in a highly programmable manner.” I suggest that the authors reformulate this, e.g. remove "highly" and avoid repeating “programmable manner”.

Response: We have rephrased the corresponding two sentences removing repetitive "programmable manner" as follows: "Control of programmable supramolecular interactions among self-propelled building blocks should realize "molecular swarming". We demonstrate, in this study, that DNA can be employed as a universal interface to control swarming of MTs in a programmable manner."

Comment 5: -p. 6, line.126. “..output controlled..” I guess, “..output was controlled..”

Response: The corresponding sentence was properly revised as suggested.

Answers to the comments from Reviewer #5:

Comment: In the present manuscript the authors present that the swarming behavior of microtubules can be controlled by DNA hybridization. They put the ability of switching the interactions into a framework of robotics and computing. This is a rather unusual perspective – and somehow lacks the punch to really convince. I find the introduction a bit disturbingly full with buzz words.

Response: While we agree that the work could be framed purely from the perspective of a soft matter scientist, we also believe that an interdisciplinary community of researchers is emerging which is emphasizing the connection between these molecular systems and classical problems in the engineering of robots and autonomous systems. This emerging research field called "molecular robotics", in which functional molecules are integrated into a system composed of three essential factors comprising "robots" (sensors, information processors, and actuators) is outlined in ref. 11 (Hagiya et al.: Molecular robots with sensors and intelligence. *Acc. Chem. Res.* **47**, 1681-1690 (2014)). This perspective is written by researchers who include a chemist, a biologist, a roboticist,

a computer scientist and a mathematician. We agree that the concept seems "unusual" at this moment, the concept is used in recent studies published in high impact journals: i.e. ref. 14 (Kassem et al.: Stereodivergent synthesis with a programmable molecular machine. *Nature* **549**, 374 (2017)) or ref. 13 (Sato et al.: Micrometer-sized molecular robot changes its shape in response to signal molecules. *Science Robotics* **2**, eaa13735 (2017)). We believe that the framework will be widely accepted in the near future, and that the present study will be recognized as a valuable contribution. For a journal aiming at a general audience, such as *Nature Communications*, communicating with readers who are not only chemists requires authors to make connections to other fields and take a broad perspective. These connections can then come across as gratuitous buzzwords, but we strongly believe that this can entice readers to take a closer look at a work which they may not have considered otherwise.

Following the reviewer's advice, we have combed over the introduction and removed unnecessary buzzwords (replacing "rational control" with "control" and "precise construction" with "construction").